# Effects of Soluble Guanylate Cyclase Stimulators and Activators on Anti-Aggregatory Signalling in Patients with Coronary Artery Spasm

**DOI:** 10.3390/ijms24119273

**Published:** 2023-05-25

**Authors:** Armin Muminovic, Yuliy Y. Chirkov, John D. Horowitz

**Affiliations:** Basil Hetzel Institute for Translational Research, University of Adelaide, 37a Woodville Road, Adelaide, SA 5011, Australia; armin.muminovic@flinders.edu.au (A.M.); yuliy.chirkov@adelaide.edu.au (Y.Y.C.)

**Keywords:** nitric oxide, soluble guanylate cyclase, riociguat, cinaciguat, platelet aggregation, coronary artery spasm

## Abstract

Impairment of the nitric oxide/soluble guanylate cyclase (NO)/sGC) signalling cascade is associated with many forms of cardiovascular disease, resulting not only in compromised vasodilatation but also loss of anti-aggregatory homeostasis. Myocardial ischaemia, heart failure, and atrial fibrillation are associated with moderate impairment of NO/sGC signalling, and we have recently demonstrated that coronary artery spasm (CAS) is engendered by severe impairment of platelet NO/sGC activity resulting in combined platelet and vascular endothelial damage. We therefore sought to determine whether sGC stimulators or activators might normalise NO/sGC homeostasis in platelets. ADP-induced platelet aggregation and its inhibition by the NO donor sodium nitroprusside (SNP), the sGC stimulator riociguat (RIO), and the sCG activator cinaciguat (CINA) alone or in addition to SNP were quantitated. Three groups of individuals were compared: normal subjects (*n* = 9), patients (Group 1) with myocardial ischaemia, heart failure and/or atrial fibrillation (*n* = 30), and patients (Group 2) in the chronic stage of CAS (*n* = 16). As expected, responses to SNP were impaired (*p* = 0.02) in patients versus normal subjects, with Group 2 patients most severely affected (*p* = 0.005). RIO alone exerted no anti-aggregatory effects but potentiated responses to SNP to a similar extent irrespective of baseline SNP response. CINA exerted only intrinsic anti-aggregatory effects, but the extent of these varied directly (r = 0.54; *p* = 0.0009) with individual responses to SNP. Thus, both RIO and CINA tend to normalise anti-aggregatory function in patients in whom NO/sGC signalling is impaired. The anti-aggregatory effects of RIO consist entirely of potentiation of NO, which is not selective of platelet NO resistance. However, the intrinsic anti-aggregatory effects of CINA are most marked in individuals with initially normal NO/sGC signalling, and thus their magnitude is at variance with extent of physiological impairment. These data suggest that RIO and other sGC stimulators should be evaluated for clinical utility in both prophylaxis and treatment of CAS.

## 1. Introduction

Circulatory homeostasis, at the level of both regulatory haemostasis and vascular integrity, relies on the stability of platelet–vessel wall interactions and therefore is ultimately subject to maintenance of both normal vascular and platelet reactivity. In recent years, there has been increasing emphasis on structural and physiological impairment of macrovascular function as potentially dominant components of disturbances of circulatory homeostasis [1]. For example, the classical findings of Furchgott and Zawadzski in intact or de-endothelialised isolated perfused vessels [2] gave rise to the concept that “endothelial dysfunction” might be a central cause of not only impaired vascular reactivity, but also propensity towards atherogenesis and development of mural thrombosis [3,4]. In time, it was recognised that one critically important endothelial-derived vasodilator and anti-aggregatory agent was nitric oxide (NO) [1,5]. Although it is now clear that the vascular endothelium generates a myriad of vasoactive substances other than NO [1], subsequent research has tended to focus on the pathophysiology of both NO generation and signalling. Although generation of NO may be impaired in a variety of cardiovascular disease states, largely reflecting impairment of NO synthase function [6,7], it has been shown that NO *signalling* is also impaired (“NO resistance”) in patients with many forms of cardiovascular disease [8,9,10].

Critically, it has been known for many years that NO is a pivotally important modulator of platelets and vascular homeostasis, exerting substantial anti-aggregatory effects [4,11,12]. Thus, it is apparent that NO resistance may theoretically increase risk not only of vasoconstriction but also of dysfunctional platelet–vascular interactions. We recently documented that patients with coronary artery spasm (CAS), a very common but frequently misdiagnosed cause of cardiovascular morbidity and mortality [13] notorious for poor anti-anginal response to NO donors [14], exhibit particularly severe NO resistance at the level of platelet aggregation [15]. Furthermore, we showed that symptomatic crises in CAS patients are engendered by mast cell-derived inflammatory activation, culminating in vascular, endothelial, and platelet damage, and aggravation of NO resistance at the platelet level [15]. In comparison, platelet NO resistance in patients with stable angina pectoris is readily detectable but much less severe than that seen in CAS [16]. 

One of the major mechanisms implicit in the pathogenesis of NO resistance at the platelet level is impaired function of the “receptor” for NO, the heme-protein soluble guanylate cyclase (sGC) [10,17]. Conversely, it has been shown that it is possible to reverse the physiological consequences of dysfunction of the sGC protein via one of two approaches, both developed approximately twenty years ago by Bayer pharmaceuticals. These approaches constituted “activation” or “stimulation” of sGC [18]. In particular, sGC stimulators have been found to be clinically useful in improving the symptomatic status and prognosis in patients with pulmonary arterial hypertension (PAH) [19]. Interestingly, these agents represent the only known effective treatment for chronic thrombo-embolic pulmonary hypertension (CTEPH), where PAH develops as a long-term complication after a pulmonary embolism [19].

Considering these data, we wondered whether either sGC stimulators or activators might prove effective in CAS, on the basis that this is a condition associated both with extreme NO resistance, and which, like CTEPH, pivotally involves both vascular smooth muscle and platelets [15]. We therefore designed a study comparing the effects of the sGC activator cinaciguat (CINA) and those of the sGC stimulator riociguat (RIO) in patients with proven CAS, studied during the chronic symptomatic phase. In order to evaluate potential heterogeneity of effects of CINA and RIO, we assessed the impact of these agents on platelet sGC signalling across the entire spectrum of baseline sGC responsiveness by including not only a normal control group but also patient comparator groups consisting of patients with different cardiovascular disease states, who were anticipated to have a lesser degree of NO resistance than CAS patients [10,15].

## 2. Methods

### 2.1. Patients and Control Subjects: Selection

The study compared (a) normal adult subjects of both sexes with no known medical conditions, cardiovascular or otherwise, known to induce NO resistance with (b) Group 1 [mild/moderate NO resistance] patients (again of both sexes, plus anticipated to have mild to moderate NO resistance on the basis of at least one of the following: stable myocardial ischemia, heart failure, hypertension, atrial fibrillation, diabetes mellitus and/or previous myocardial infarct) [9,10,16] and (c) Group 2 [CAS] patients (anticipated to have *severe* NO resistance). 

All patients were recruited through the QE Specialist Centre, Woodville, South Australia, Australia. Patients with previously diagnosed CAS, either predominantly involving large vessels (Prinzmetal Angina: PA) or small coronaries (Coronary Slow Flow Phenomenon: CSFP [20]), were recruited. In all cases, diagnosis had been previously confirmed at coronary angiography, which showed the absence of haemodynamically significant stenoses of the epicardial coronary arteries plus either:(a)Inducible coronary artery spasm via intracoronary injection of 25 to 100 µg acetylcholine (ACh) resulting in a >75% reduction in luminal diameter (focal or diffuse) together with onset of chest pain [21] OR.(b)Presence of CSFP, defined as TIMI-2 flow in at least one major epicardial vessel [20].

Ethical approval for the study was granted from the TQEH/LMH/MH Human Research Ethics Committee and Central Adelaide CALHN Research Governance (approval no. HREC/15/TQEH/75). All human studies were performed in accordance with the declaration of Helsinki, and informed consent was obtained from all subjects. 

### 2.2. Platelet Aggregometry

Venous blood (10 mls) was taken via venepuncture of an antecubital vein. Blood was collected using tubes containing a 1:10 volume of acid citrate anticoagulant. The extent of platelet aggregation was analysed in whole blood using a 4-channel impedance aggregometer (Model 700, Chrono-log, Havertown, PA, USA) according to the manufacturer’s specifications. The experimental protocol applied was exactly as described by us previously [15,16].

Briefly, tests were performed on 450 µL of blood diluted with 500 µL of saline at 37 °C with a stirring speed of 900 rpm. The protocol involved the insertion of an electrode containing a pair of palladium-made wires directly into the continuously stirring blood. The extent of platelet aggregation between the wires is represented as a measure of increased electrical resistance in Ohms (Ω). 

Platelet aggregation was induced by ADP (2.5 µM). Serial dilutions of CINA and RIO were added to test cells at 5 and 2 min prior to addition of ADP, respectively. The NO donor sodium nitroprusside (SNP: 10 µM) was incubated in test cells for 1 min prior to ADP. Dimethyl sulfoxide (DMSO) was used as the vehicle for both RIO and CINA. Initial experiments were performed to exclude intrinsic anti-aggregatory effects of DMSO. Aggregation was monitored continually for 7 min, and responses were displayed as electrical impedance in ohms. 

Initial experiments involved the utilisation of up to 5 concentrations of RIO and CINA varying from 0.01 µM to 10.0 µM to determine concentrations which would elicit substantial, but not maximal, inhibitory responses. These concentrations were to be used in the final protocol. A total of 17 patients were used for pilot RIO experiments with a mean logEC50 of −5.727 M. A total of 14 patients were used for pilot CINA experiments with a mean logEC50 of −5.747 M. Thus, the 1 µM concentration was established as an approximate measure of EC50 concentrations for both RIO and CINA. 

### 2.3. Statistical Methodology

Anti-aggregatory effects of SNP for the control versus patient groups (as an agglomerate) were analysed by Student’s non-paired *t*-test. Comparison between the heterogenous/moderate group and patients with CAS was analysed by ANOVA followed by Dunnett’s post hoc *t*-test. Student’s paired *t*-test was used to analyse inhibitory effects of RIO and CINA in isolation. Inhibition of platelet aggregation by RIO with SNP and CINA alone was analysed in control subjects and CAS patients again using Student’s paired *t*-test. Pearson’s correlation coefficients were utilised to assess the relationship between basal SNP responsiveness and RIO/CINA anti-aggregatory responses. Data are expressed as mean ± SEM for normally distributed data. Proportional data were analysed using either χ2 or Fisher’s exact test depending on compartmental sizes. Differences in age between groups were analysed using an unpaired *t*-test. The level of statistical significance was determined as *p* < 0.05 throughout. All analyses were performed using GraphPad Prism 9. 

## 3. Results

### 3.1. Subject/Patient Characteristics

The characteristics of control subjects and patients are described in Table 1. Control subjects were substantially younger than patients. Two of the control subjects were undergoing treatment for borderline hypertension and one had suffered a single episode of paroxysmal atrial fibrillation. Otherwise, they had no known cardiovascular disease states. 

Patients in Group 1 were significantly older than those in Group 2, and many had haemodynamically significant large coronary artery stenoses. Almost half of these patients had either paroxysmal or permanent atrial fibrillation. Conversely, many were receiving drugs which are known to limit NO resistance, such as ACE-inhibitors for management of hypertension, heart failure, or coronary risk [22], perhexiline for stable severe angina pectoris [23], and statins for management of hypercholesterolaemia [24]. All patients receiving perhexiline were also receiving statin therapy. It was anticipated that the admixture of factors inducing and ameliorating NO resistance among the Group 1 patients would result in the requisite broad range of platelet responsiveness to NO.

Group 2 patients (those with CAS) were all studied during the chronic phase of the disease. Of the 16 CAS patients, 8 had PA and 8 had CSFP. The majority of these patients were studied while receiving treatment with L-type calcium channel antagonists as prophylaxis against angina symptoms. 

### 3.2. Anti-Aggregatory Effects of the NO Donor SNP

These data are depicted in Figure 1.

Anti-aggregatory effects of SNP were significantly less for patients than for controls (Student’s non-paired *t*-test, *p* = 0.021). As envisaged, the three groups studied represented a spectrum of platelet anti-aggregatory responsiveness to SNP. Mean responses were 30 ± 5.1% for control subjects, 19.8 ± 3.6% for Group 1 patients, and 3.4 ± 2.9% for Group 2 patients. Post hoc comparison of the two patient groups (Dunnett’s *t*-test) demonstrated more severe impairment of SNP responses in Group 2 (CAS) patients (*p* = 0.0054). Interestingly, many Group 2 patients exhibited paradoxical mild pro-aggregatory responses to SNP. 

### 3.3. Intrinsic Anti-Aggregatory Effects of RIO and CINA

In order to determine whether either RIO or CINA might exert intrinsic anti-aggregatory effects, experiments were performed with their incubation with blood from patients. RIO (1 µM) alone induced no change in the extent of ADP-induced aggregation. (Figure 2) Conversely, addition of CINA (1 µM) alone induced a mean 37% reduction in extent of ADP-induced aggregation (*n* = 8, *p* = 0.0001). 

### 3.4. Potentiation of Anti-Aggregatory Responses to NO by RIO Is Independent of NO Resistance

Unlike CINA, RIO exhibited no intrinsic anti-aggregatory effects (Figure 2). We next investigated whether RIO, as an sGC stimulator, potentiated the anti-aggregatory effects of SNP (and thus of NO). Experiments were performed across the entire spectrum of subjects and patients.

In the case of control subjects (Figure 3A), addition of RIO to SNP resulted in potentiated inhibition of ADP-induced aggregation from an initial mean SNP response of 34 ± 4.5% to an 83 ± 4.0% response post RIO (*p* < 0.0001). In Group 2 patients (CAS), RIO induced a similar extent of potentiation from an initial mean SNP response of 5 ± 3.3% to 45 ± 6.9% (*p* < 0.0001). We then utilised the entire control/patient spectrum of anti-aggregatory effects of NO in order to determine whether incremental effects of RIO were independent of baseline responsiveness to NO: as shown in Figure 3B, potentiation of anti-aggregatory responses to SNP was independent of initial response.

### 3.5. Inhibition of ADP-Induced Aggregation by CINA Is Directly Proportional to Platelet Responsiveness to NO

Unlike RIO, CINA exhibited anti-aggregatory effects in the absence of SNP (Figure 2).

Nevertheless, we sought to determine whether the extent of anti-aggregatory response to CINA was independent of tissue responsiveness to NO. Pooled results of individual experiments are shown in Figure 4A. 

Responses to CINA were significantly greater (*p* < 0.01 for both) than those to SNP for both normal subjects and Group 2 (CAS) patients. When responses to CINA were correlated with those to SNP across the entire normal subject/patient spectrum, as shown in Figure 4B, there was a highly significant direct correlation between SNP and CINA responses. Thus CINA, while not interacting directly with platelet responsiveness to NO donors, activates normally functioning sGC more effectively than dysfunctional enzymes.

### 3.6. Anti-Aggregatory Responses to RIO Are Not Subject to Significant Physiological Antagonism

It would be expected that individual patient responses to all anti-aggregatory agents would, to some extent, be subject to principles of physiological antagonism [25,26] and therefore tend to vary inversely with extent of ADP-induced aggregation. In the currently examined entire data set, ADP-induced aggregation did not vary significantly (ANOVA) between the control and two patient groups. When the relationships between extent of response to ADP and of inhibition of aggregation were considered (Figure 5), neither SNP alone (r= −0.012, *p* = NS) nor the incremental effect of RIO over SNP (r= −0.28, *p* = NS) showed evidence of substantial physiological antagonism.

## 4. Discussion

Many forms of cardiovascular disease are pivotally associated with impairment of the NO/sGC pathway [8,9,10]. These include both heart failure and myocardial ischaemia, affecting predominantly the left side of the heart, and pulmonary hypertension [27], affecting the right side of the heart. there is ample evidence that amelioration of function of this cascade is associated with improved outcomes. Recently, a new approach to the problem of impaired sGC signalling has become available with a development of sGC stimulators and activators. However, this group of drugs has thus far been utilised clinically almost entirely in the management of pulmonary hypertension [19,27].

We have recently reported that coronary artery spasm (CAS), whether engendered primarily by large or small coronary constriction, represents an example of profound, although fluctuating, impairment of the NO/sGC pathway [15]. Clinically, quality of life for many patients with CAS remains poor despite the widespread availability of L-channel calcium antagonists. Indeed, crises in CAS patients represent an increasingly common basis for presentation to emergency departments [14]. We therefore sought to determine whether the spectrum of utility of sGC stimulators and activators might potentially extend to the prophylaxis of treatment of CAS. Given that attacks of CAS, while precipitated by mast cell activation, pivotally involve incremental platelet–endothelial interactions, with both activation and fragmentation of platelets [15] the current “proof of principle’’ experiments were focused on limitation of platelet aggregability by sGC activators and stimulators. We chose to study RIO as a clinically applicable sGC stimulator and CINA as an sGC activator. We also chose to compare the effects of these two drugs in three groups of individuals: normal subjects, Group 1 patients (in whom mild impairment of NO/sGC signalling was to be expected) [8,9], and in Group 2 patients with CAS (in whom our previous data led us to expect that NO/sGC signalling would be severely impaired) [15]. 

As expected, these three groups of subjects/patients showed progressive diminution in anti-aggregatory responses to the NO donor SNP. Indeed, in patients with CAS, SNP frequently induced small paradoxical pro-aggregatory responses. These may represent the results of biased sGC signalling in CAS patients, with a consequent shift to generation of cyclic inosine monophosphate (cIMP) rather than cGMP, as previously described in vascular smooth muscle in the presence of hypoxia [28]. Categorically speaking, the effects of RIO and CINA were as expected: RIO was inactive by itself but markedly potentiated the effects of SNP, while the effects of CINA were entirely independent of presence/absence of a NO donor. 

The startling result was that RIO restored NO responsiveness to normal in all groups investigated without exhibiting selective hyper-responsiveness among Group 2 patients and without being significantly subject to physiological antagonism. On the other hand, the intrinsic anti-aggregatory effects of CINA were directly proportional to patients’ baseline responsiveness to NO: thus, CINA produced maximal anti-aggregatory effects which were least marked for patients in Group 2. Thus, in theory, CINA would most markedly potentiate sGC signalling in normal subjects rather than patients, while RIO would tend to normalise NO/sGC signalling in patients with severe NO resistance, such as those with CAS.

The current study has a number of potential limitations. First of all, all of the CAS patients were studied electively rather than during a crisis, during which anti-aggregatory responses to NO would be minimal [15]. Therefore, we do not know precisely whether NO responsiveness would have been completely restored in this context. Furthermore, it was not possible to investigate effects of systemically administered RIO; therefore, we have no data regarding either changes in vascular reactivity or the risk of clinically significant hypotension, which has been a problem in clinical trials of sGC stimulators [19,29,30]. Furthermore in vivo administration of RIO would have enabled us to obtain data on the impact of glycocalyx shedding, a major problem in CAS crises [15], and on fragmentation of platelets, which also occurs during these crises. Similarly, in vivo administration of the drug to a substantial number of patients would be necessary for clinical dose ranging and for determination of bleeding risk. 

Other agents currently used in prophylaxis or treatment of CAS include nitroglycerine, L-channel calcium antagonists, corticosteroids, and the hydrogen sulphide donor NAC. Most importantly, we did not extensively explore interactions with hydrogen sulphide donors but would predict some sort of incremental effect. Furthermore, a number of recent investigations have suggested that disturbances of microvascular reactivity may be of considerable importance in the pathogenesis both of stable angina pectoris [29] and of heart failure [31]. This provides an incremental theoretical rationale for more extensive inception of the clinical use of sGC stimulators in such contexts. Indeed, the sGC stimulator vericiguat has recently been reported to be effective in the clinical context of heart failure, reducing the rise of the combination of cardiovascular death and hospital admission for heart failure with no significant increase in associated major bleeding risk [32]. Furthermore, our current data potentially carry major implications as regards mechanisms of beneficial effects of sGC stimulators in patients with PAH secondary to CTEPH. In this respect, recent experimental data suggest that sGC stimulators may partially reverse increases in pulmonary vascular resistance (and potentially also reverse ongoing thrombosis) via enhanced autacoidal signalling [33,34]. Unfortunately, the concept of resistance to homeostatic autacoidal effects has received little previous attention in PAH; this represents a desirable specific focus for future studies.

Overall, the current data suggest that sGC stimulators, such as RIO, should also be evaluated clinically as components of medical management of patients with CAS (as exemplified by the majority of Group 1 patients in the current study) where many patients remain symptomatic despite utilization of calcium antagonists and long-acting nitrates. Indeed, there is a recent single case report in the literature concerning a patient with a refractory CAS crisis [35]. Translation of these theoretical advantages to controlled clinical trial evaluation is therefore eagerly awaited.

## Figures and Tables

**Figure 1 ijms-24-09273-f001:**
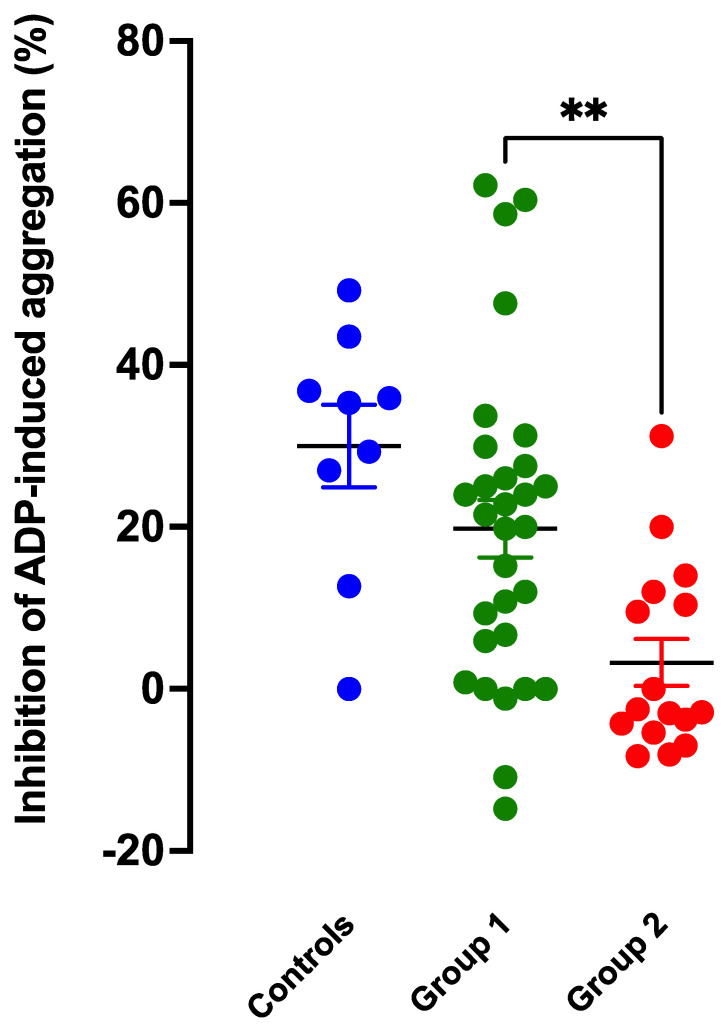
Inhibition of ADP-induced platelet aggregation by SNP (10 µM) in whole blood samples from control subjects (*n* = 9; blue) and patients of Group 1 (*n* = 30; green) and Group 2 (*n* = 16; red). Patients in Group 1 had risk factors (other than CAS) for NO resistance. All patients in Group 2 had CAS. ** = *p* < 0.01 for difference.

**Figure 2 ijms-24-09273-f002:**
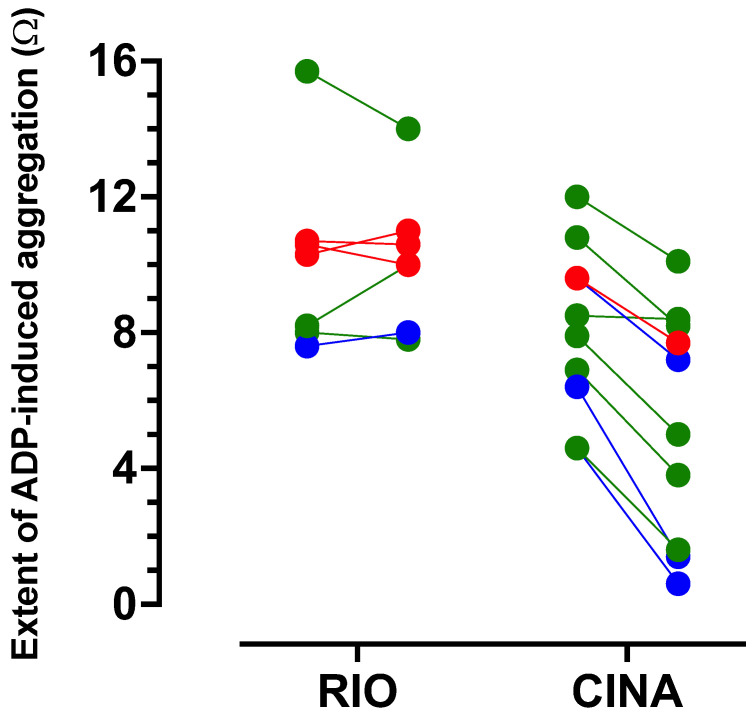
The extent of ADP induced aggregation (Ω) in blood pre and post addition of RIO alone (1 µM, *n* = 7) and CINA alone (1 µM, *n* = 20) in control (blue) and Group 1 (green) and Group 2 (red) patients.

**Figure 3 ijms-24-09273-f003:**
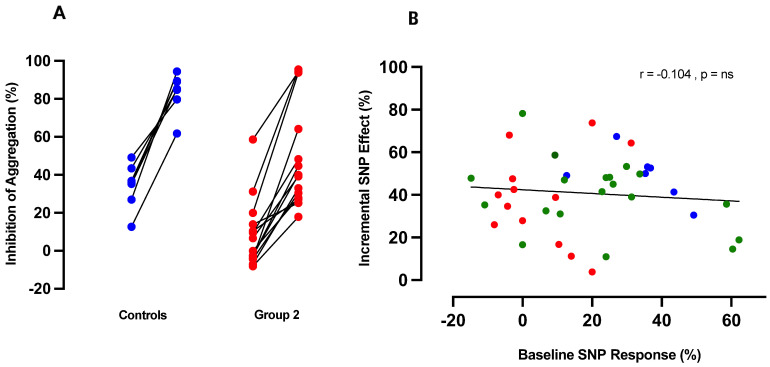
(**A**) Inhibition of ADP-induced platelet aggregation before and after addition of RIO (1 µM) to SNP (10 µM) in whole blood samples from healthy control subjects (*n* = 6; blue) and patient Group 2 (CAS) (*n* = 13, red). (**B**) Correlation of the effects of RIO (1 µM) with initial anti-aggregatory response to SNP (10 µM) in control (blue), patient Group 1 (green), and patient Group 2 (red): *n* = 40.

**Figure 4 ijms-24-09273-f004:**
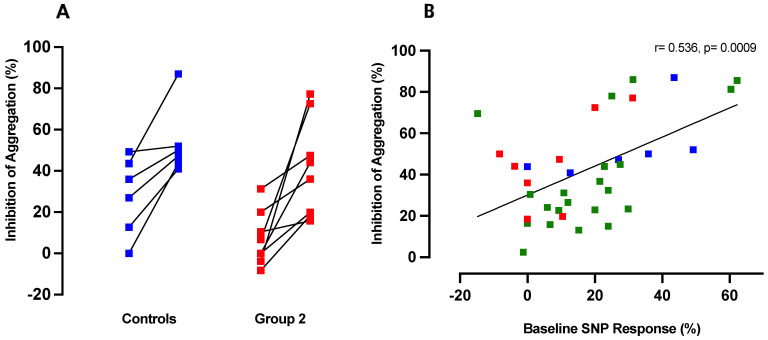
(**A**) Inhibition of ADP-induced platelet aggregation by addition of CINA (1 µM) alone in whole blood samples from healthy control subjects (*n* = 6; blue) and Group 2 (CAS) patients (red; *n* = 9). (**B**) Comparison of the anti-aggregatory effects of CINA (1 µM) with those of the NO donor SNP (10 µM) in control (blue), patient Group 1 (green), and patient Group 2 (red): *n* = 35.

**Figure 5 ijms-24-09273-f005:**
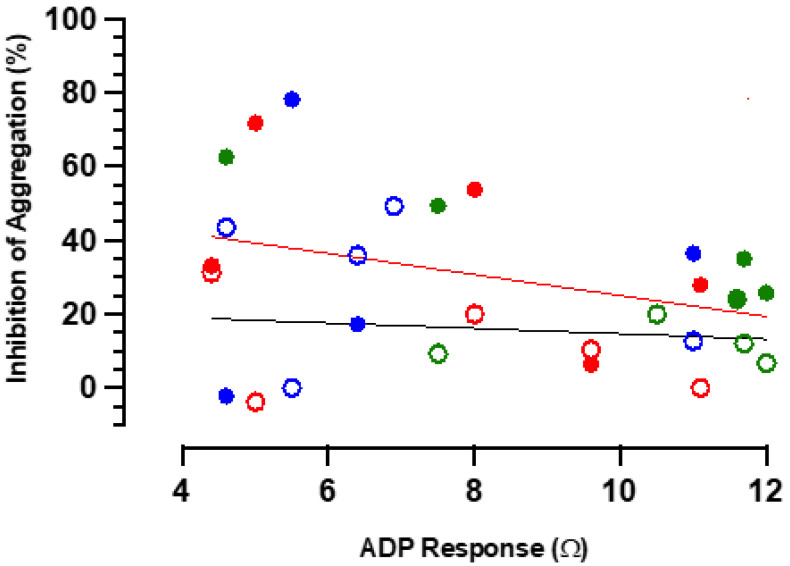
Relationships between the extent of ADP-induced platelet aggregation and inhibition of aggregation by SNP alone (open symbols, black line) and incremental response to RIO (closed symbols, red line) in control (blue), patient Group 1 (green), and patient Group 2 (red); *n* = 13.

**Table 1 ijms-24-09273-t001:** Comparison of patient characteristics across the three subject/patient groups. All statistical comparisons were between patient groups 1 and 2.

Characteristic	Controls (*n* = 9)	Patients: Group 1 (*n* = 30)	Patients: Group 2 (*n* = 16)	*p*
Men/Women, *n*	4 vs. 5	18 vs. 10	4 vs. 14	0.007 *
Age (mean/SEM)	56 ± 8	74 ± 2	64 ± 3	0.006 ^#^
Coronary Stenoses, *n* (%)	0	10 (33)	0	0.008 *
Heart Failure, *n* (%)	0	7 (23)	0	0.071
AF, *n* (%)	1 (11%)	11 (36)	1 (6)	0.035 *
Diabetes, *n* (%)	0	14 (46)	2 (12)	0.026 *
Hypertension, *n* (%)	2 *	13 (43)	2 (12)	0.049 *
Medication				
Aspirin, *n* (%)	0	2 (6)	1 (6)	1.000
Calcium Channel Antagonists, *n* (%)	0	5 (16)	9 (56)	0.008 *
NAC, *n* (%)	0	0 (0)	4 (25)	0.011 *
ACE inhibitors, *n* (%)	2 (22)	6 (20)	2 (12)	0.694
Perhexiline, *n (%)*	0	9 (30)	1 (6)	0.130
Statins, *n* (%)	1	8 (26)	2 (12)	0.455

SEM = standard error of mean, AF = atrial fibrillation, NAC = N-acetylcysteine. ACE = angiotensin converting enzyme. * Fishers’s exact test. ^#^ Student’s unpaired *t*-test.

## Data Availability

All of the data concerning individual or group data published within this report are freely available on reasonable request.

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
