# Peer review of "Effects of Soluble Guanylate Cyclase Stimulators and Activators on Anti-Aggregatory Signalling in Patients with Coronary Artery Spasm"

_ijms, 2023, doi:10.3390/ijms24119273_

Round 1
Reviewer 1 Report
Werey well written, prese
Author Response
We greatly appreciate the Reviewer's endorsement of our original manuscript.
Reviewer 2 Report
*X/C/V
Research on the influence of signaling factors on contractility-vasodilatation homeostasis, as well as anti- and pro-aggregation homeostasis, are very nicely presented. This primarily refers to disorders of the function of coronary blood vessels, but at the same time it is also an inpentive for the creation of new RIO drugs, in place of antagonism of calcium and inhibiting nitrates and other vasodilators.
We advise you to include in the discussion some very important findings in research on the development of pulmonary hypertension due to hitherto irreversible changes in PAs. Newer research in this area predicts the possibility of reversing severe capillary changes in PAG primarily by stopping the depression of apoptosis, clarifying the function of pericytes and anti-inflammatory drugs. I recommend that at least with a few references you refer to the similarity of the activity on the coronary blood vessels and the distal pulmonary blood vessels (how to stop self-advancement - self perpetuation) and inflammation, that is, how to find stimulating factors.
Author Response
We appreciate the Reviewer's constructive suggestions. While it is true that the primary thrust of our experiments involved investigation of the potential utility of sGC stimulators (and activators) for pro-thrombotic disorders within the systemic circulation, and especially in patients with coronary artery spasm, it is also important to reflect that our results may have additional implications for the beneficial effects of sGC stimulators and/or activators in patients with Pulmonary Artery Hypertension (PAH), especially CTEPH. Indeed, as the Reviewer points out, the beneficial effects of these agents may now be recognised to include partial reversal of apparently "fixed" increases in resistance of pulmonary microvessels. We have added comments to this effect within the Discussion of the revised manuscript, emphasising the re-interpretation of drug effect and including 2 new references which ultimately relate to the pro-homeostatic role of sGCstimulation especially at the level of the platelet/microvascular interface in such patients.
Reviewer 3 Report
Conguratulations for your work. I have some suggestions:
1. The inclusion and exclusion criteria are rather difficult to follow, show with a flowchart
2. Please clearly define health subjects, how are you sure they are healthy, there are old people and they are expected to have comorbidities. How well do you know their medical records/history. Please define in detail
3. How much in mL venous blood did you need for the study for each individual?
4. results: "Conversely, many were re- 164 ceiving drugs which are known to limit NO resistance, such as ACE-inhibitor [22], perhex- 165 iline [23] and statins [24]" define this situation more clearly. How many patients use these drugs, is there any one to use multiple drugs amongst these ones, how this situation implied your research. Plese give precise neumbers and revise the materials and methods section accordingly
5. Discussion mainly consisted of the previous data of the authors. Discuss different studies, the results must be compared with the previous studies of other authors.
6. Discussion: Please give more detail about the potential clinical impacts.
Author Response
We appreciate the Reviewer's compliments re our overall manuscript, and even more so his/her helpful suggestions.To summarise the changes made:-
(1) Clarify the inclusion and exclusion criteria for the various groups. Please use a flow-chart. We agree that a major problem is imposed by the heterogeneity of selection criteria for Group 1 Patients (individuals with presumptive mild/moderate NO resistance). The basis for the requirement of such a heterogeneous group was our intent to evaluate impact of sGC stimulation or activation across the entire spectrum of platelet responsiveness to NO, as depicted in Figures 3B and 4B. We have added clarification details, especially regarding Group 1, to the text , and hope that these additional details will resolve any potential confusion in readers.
(2) Please define healthy subjects... We agree that elderly patients almost always have some comorbidities, and these are often cardiac. In the revised text we have defined inclusion criteria for these "normal subjects", and indeed have also commented on the fact that they were substantially younger than the majority of patients in Group 1.
(3) How much venous blood is required? A total of 10 mls is required: this is now indicated in the text.
(4) Line 164 onwards: "Many were receiving drugs known to limit NO resistance.." The numbers of involved patients, almost all of whom were in group 1, is indicated in Table 1. We have added additional details in the text regarding the use of more than one of these agents in some patients. We have also included this issue in the Patient Selection section, as suggested by the reviewer.
(5) Cite relevant work of other authors in Discussion: We have no wish to inappropriately self-citate! Unfortunately, the literature is largely devoid of work on NO resistance apart from our own studies. In order to be completely rigorous about this, we performed a literature search in the context of PAH. Unfortunately, the concept that responsiveness of platelets to NO might be impaired in patients with PAH has not yet been investigated! Outside the specific context of PAH, there are some previous publications, especially from one Italian group. This issue is now mentioned in Discussion.
(6) Give more detail about potential clinical impact: We thank the Reviewer for this suggestion and have now expanded the area in the Discussion.
Round 2
Reviewer 3 Report
The recommendations were mostly performed thank you